# Activity Guided Isolation of Phenolic Compositions from *Anneslea fragrans* Wall. and Their Cytoprotective Effect against Hydrogen Peroxide Induced Oxidative Stress in HepG2 Cells

**DOI:** 10.3390/molecules26123690

**Published:** 2021-06-17

**Authors:** Shuyue He, Xiaoyan Cui, Afsar Khan, Yaping Liu, Yudan Wang, Qimin Cui, Tianrui Zhao, Jianxin Cao, Guiguang Cheng

**Affiliations:** 1Faculty of Agriculture and Food, Kunming University of Science and Technology, Kunming 650500, China; kmlgdx303@163.com (S.H.); liuyaping@mail.kib.ac.cn (Y.L.); sdlcwyd@163.com (Y.W.); 18908742179@163.com (Q.C.); food363@163.com (T.Z.); jxcao321@hotmail.com (J.C.); 2Foundation Department, Hai Yuan College, Kunming Medical University, Kunming 650106, China; cuixiaoyan910@163.com; 3Department of Chemistry, Abbottabad Campus, COMSATS University Islamabad, Abbottabad 22060, Pakistan; afsarhej@gmail.com; 4National and Local Joint Engineering Research Center for Green Preparation Technology of Biobased Materials, Yunnan Minzu University, Kunming 650500, China

**Keywords:** *Anneslea fragrans*, antioxidant, guided isolation, oxidative stress, flavonoid glycosides

## Abstract

*Anneslea fragrans* Wall., commonly known as “Pangpo Tea”, is traditionally used as a folk medicine and healthy tea for the treatment of liver and intestine diseases. The aim of this study was to purify the antioxidative and cytoprotective polyphenols from *A. fragrans* leaves. After fractionation with polar and nonpolar organic solvents, the fractions of aqueous ethanol extract were evaluated for their total phenolic (TPC) and flavonoid contents (TFC) and antioxidant activities (DPPH, ABTS, and FRAP assays). The *n*-butanol fraction (BF) showed the highest TPC and TFC with the strongest antioxidant activity. The bio-guided chromatography of BF led to the purification of six flavonoids (**1**–**6**) and one benzoquinolethanoid (**7**). The structures of these compounds were determined by NMR and MS techniques. Compound **6** had the strongest antioxidant capacity, which was followed by **5** and **2**. The protective effect of the isolated compounds on hydrogen peroxide (H_2_O_2_)-induced oxidative stress in HepG2 cells revealed that the compounds **5** and **6** exhibited better protective effects by inhibiting ROS productions, having no significant difference with vitamin C (*p* > 0.05), whereas **6** showed the best anti-apoptosis activity. The results suggest that *A. fragrans* could serve as a valuable antioxidant phytochemical source for developing functional food and health nutraceutical products.

## 1. Introduction

Reactive oxygen species (ROS) are produced as by-products though oxidative respiration in normal physiological and biochemical processes [1]. As endogenous free radicals, ROS plays an important role in cell signaling and the maintenance of body constancy in a normal range [2]. If the ROS could not be effectively scavenged, excessive ROS could lead to the occurrence of oxidative stress, which may influence cell proliferation and apoptosis [3]. Oxidative stress is closely related to a variety of diseases including diabetes, hyperlipemia, obesity, cancer, and cardiovascular and neurodegenerative diseases [4]. Recently, many studies have evidenced that a diet enriched with antioxidants possesses a series of beneficial effects owing to their scavenging ability on excessive ROS [5,6].

Natural antioxidants contain a variety of molecules such as polyphenols [7], carotenoids [8], vitamins [9], nitrogen-containing compounds [10], and coumarins [9]. Polyphenols are distributed widely throughout the plant kingdom and promote health benefits owing to their antioxidant properties [11]. Many of them, such as resveratrol and some derivatives [12], anthocyanidins [13], isoflavones [14], catechin [15], and quercetin [16] are well known for their protective effect by scavenging ROS [1]. For the past few years, polyphenols have been attracting attention in the prevention of cancer [17], cardiovascular dysfunction [18], neurodegenerative diseases [19], and aging [20]. Therefore, searching for effective antioxidants is an urgent need to promote human health.

*Anneslea fragrans* Wall. is an evergreen plant, which is mainly distributed in the southern of China [21]. In addition to its ornamental purpose, the leaves of *A. fragrans* are also used as a folk medicine to treat fever, liver protection, invigorating stomach and intestines in China and Cambodia [22], which had been recorded in “Yunnan Simao Chinese Herbal Medicine”. In addition, the leaves have also been processed as a tea beverage, which is known as “Pangpo Tea”. In previous reports [23], the extract of *A. fragrans* has shown antioxidant and antimalarial activities. However, to date, its antioxidant phenolic compounds from *A. fragrans* have not been reported. 

Thus, the purpose of this research was to isolate and identify the antioxidant compounds from *A. fragrans* leaves that are responsible for its traditional use for the treatment of liver diseases. The four fractions, dichloromethane fraction (DF), ethyl acetate fraction (EAF), *n*-butanol fraction (BF), and residual water fraction (RWF) of aqueous ethanol extract were assessed for their total phenolic (TPC) and total flavonoid contents (TFC) and for their antioxidant capacity. The *n*-butanol fraction (BF) had the highest TPC and TFC with strongest antioxidant activity. The bio-guided fractionation of BF allowed the purification of compounds. Furthermore, the cytoprotective effect of the isolated compounds was performed on hydrogen peroxide (H_2_O_2_)-induced oxidative stress in human liver cancer HepG2 cells. The intracellular ROS production and cell apoptosis were determined using flow cytometry. Thus, this research afforded a valuable antioxidant phytochemical ingredient for the development and utilization of *A. fragrans* leaves as a functional supplement (healthy tea) in food and health industry. 

## 2. Results and Discussion

### 2.1. Yield Efficiency of Fractions and Subfractions

The liquid–liquid partitioning by organic solvents and column fractionation are important techniques for enriching the bioactive compounds from crude extract [24]. Using liquid–liquid partitioning, ethanol extract (CE) from *A. fragrans* leaves was successively fractionated with dichloromethane, ethyl acetate, and *n*-butanol to give four fractions (DF, EAF, BF, and RWF), respectively. Yield percentages of fractions were found to vary from 11 to 27%. The residual water fraction (RWF, 27%) had the highest yield, and the yields of other fractions were in the order as follows: *n*-butanol fraction (BF, 24%), ethyl acetate fraction (EAF, 21%), dichloromethane fraction (DF, 11%). 

The yield percentage of subfractions (BF-A to E) was found to vary from 3.82 to 48.4%. The BF-E presented the highest yield of 48.4% followed by BF-D (24.96%), which has significantly higher yields than the other fractions.

### 2.2. HPLC Analysis

High-performance liquid chromatography (HPLC) detection has been demonstrated to be a powerful technique for quantitative determination. HPLC analysis revealed that BF had the most compounds with the strongest antioxidant activity. Under the guidance of antioxidant assays and HPLC analysis, the antioxidative fraction was further chromatographed for the isolation of subfractions. The BF was subjected to a hydrated resin D101 column to yield five subfractions (BF-A to E). HPLC analysis revealed that BF-C to E had the most antioxidant compounds. Bio-guided fractionation of these fractions (BF-C to E) allowed the purification of seven pure compounds.

As shown in Figure 1, the BF and seven pure compounds were profiled by HPLC analysis. Based on comparison with the retention times and UV absorption curves, these seven compounds were confirmed through retention times at 7.51 min (cornoside, **7**), 7.92 min ((epi)-catechin, **6**), 9.38 min (confusoside, **1**), 9.57 min ((S)-naringenin-7-*O*-β-d-glucopyranoside, **4**), 10.32 min (vacciniifolin, **2**), 12.64 min (2′,3,4,4′-tetrahydroxydihydrochalcone, **5**), and 13.95 min (1-[4-(β-d-glucopyranosyloxy)-2-hydroxyphenyl]-3-(4-hydroxy-3-methoxyphenyl)-1-propanone, **3**) (Figure 1A). Among them, compounds **1**–**6** belong to flavonoids, and compound **7** is benzoquinolethanoid.

### 2.3. Total Phenolic Contents (TPC) and Total Flavonoid Contents (TFC)

The *A. fragrans* leaves are traditionally used as processed health tea and have been proven to be good resource of phenolics and flavonoids [25]. According to spectrophotometric assays, the TPC and TFC were tested in different fractions and subfractions from *A. fragrans* leaves. As shown in Table 1, the BF had the highest TPC value with 238.12 ± 12.05 mg GAE/g extract followed by EAF. The DF showed the lowest TPC value as 66.52 ± 0.57 mg GAE/g extract. Similarly, the highest TFC concentrations were also found in BF with TFC value as 165.19 ± 5.21 mg RE/g extract. With regard to the RWF and EAF (66.92 ± 1.38 and 116.12 ± 2.89 mg GAE/g extract, respectively), they also had lower TFC concentrations than the BF (165.19 ± 5.21 mg RE/g extract), while DF had the lowest TFC concentration as 47.94 ± 2.84 mg RE/g extract. The contents of TPC and TFC in BF were approximately four times higher than those in DF. 

The data depicted the presence of highest TPC and TFC in BF-E with values as 262.03 ± 1.72 mg GAE/g extract; 180.52 ± 6.30 mg RE/g extract, respectively, followed by BF-D (219.84 ± 4.01 and 157.01 ± 2.50 mg RE/g extract, respectively), BF-C (210.69 ± 6.31 and 151.01 ± 4.33 mg RE/g extract, respectively), while BF-B and BF-A had the lowest TPC and TFC values (Table 1). 

### 2.4. Antioxidant Activity

Most of the polyphenols, especially flavonoids and phenolic acids, are exploited into popular antioxidant foods (nutraceuticals) and present a series of human benefits [17]. In our previous study, the *A. fragrans* leaves have been proven to be a good resource of polyphenols [26]. However, the antioxidant activity of the extract from *A. fragrans* leaves and its phytochemicals have not been investigated yet. Due to different antioxidative reaction mechanisms, three assays of ABTS, DPPH, and FRAP were combined to evaluate the antioxidant activity of the fractions and subfractions from *A. fragrans* leaves.

Among the fractions, BF showed the most potent antioxidant activity in ABTS, DPPH, and FRAP radical-scavenging activities with 1808.46 ± 96.52, 951.42 ± 87.75, and 1822.96 ± 29.24 μmol TE/g extract (Table 1). DF showed the lowest antioxidative activity in ABTS and FRAP assays (139.23 ± 8.62 μmol TE/g extract and 180.05 ± 9.46 μmol TE/g extract, respectively) (Table 1). Whereas, the radical scavenging activities of four fractions (DF, EAF, BF, and RWF) in the DPPH radical-scavenging assay were found to vary from 613.38 ± 45.35 to 951.42 ± 87.75 μmol TE/g extract, respectively, having no significant difference (*p* > 0.05). These results suggested that the phytochemicals from *A. fragrans* leaves might be insensitive to DPPH. To further obtain the active metabolites, BF was selected for further fractionation. The BF was subjected to D101 macroporous adsorbing resin column chromatography eluting by a gradient of the methanol–aqueous system to yield five subfractions (BF-A to E). Hence, the ABTS radical-scavenging activity of these subfractions can be ranked as BF-E > BF-C > BF-D. The reducing activity in the FRAP assay revealed that BF-D had the strongest antioxidative activity, which was followed by BF-C and BF-E. It is noteworthy that three subfractions (BF-C to E) presented higher antioxidative activity than the mother fraction (BF). Taken together, BF-C to E were submitted to column chromatography for isolating and identifying the antioxidative phytochemicals.

### 2.5. Antioxidant Activity of the Isolated Compounds

All the isolated compounds (compounds **1**–**7**) were evaluated for antioxidant capacity by ABTS, DPPH radical scavenging activity, and FRAP antioxidant activity. All the data are described in Table 2. Compounds **6** and **2** showed the highest antioxidant activity, followed by **5**. Compound **4** displayed moderate antioxidant activity and compounds **1**, **3**, and **7** were considered inactive with ABTS and DPPH radical scavenging activities less than 200 μmol TE/g extract. Using Vc (the radical scavenging activities with 2932.91 ± 93.63 and 1873.56 ± 121.68 μmol TE/g extract, respectively) as the positive control, compound **6** had the strongest antioxidant activity in ABTS and DPPH radical scavenging assays with 1580.37 ± 89.32 and 1953.31 ± 109.93 μmol TE/g extract (*p* < 0.05), respectively. Furthermore, compound **6** had the best antioxidant capacity in the FRAP assay as 5027.43 ± 620.75 μmol TE/g extract, which was much higher than that of Vc as 3291.28 ± 241.02 μmol TE/g extract. Moreover, compounds **2** and **6** showed a significant antioxidant activity, which was equivalent to positive control (Vc). 

Additionally, the antioxidant structure–activity relationship of the flavonoids (**1**–**6**) is discussed by varying degrees of inhibitory effects. Compound **2** presented more antioxidant activity than compounds **1**, **3**, and **5**, which suggested that the 3,4-dihydroxy groups in dihydrochalcones in **2** may play a critical role against ABTS, DPPH radical scavenging activity, and FRAP antioxidant activity, and the glucose moiety substituted at C-4′ in **2** may play an important role on its antioxidant capacity [4]. Furthermore, compound **6** exhibited the highest antioxidant activity compared to other compounds, which suggested that flavan-3-ols (catechin) probably exhibit a better radical scavenging activity than dihydrochalcones and flavanones [27]. In summary, compounds **2**, **5**, and **6** with good antioxidant activity were selected further for cytoprotective effects against oxidative stress by H_2_O_2_ in the next study.

### 2.6. Relationship between Antioxidant Activity and TPC/TFC Contents

Concentrations of the TPC and TFC highly correlated with antioxidant activity from the values of FRAP (r = 0.982 and 0.977) and ABTS (r = 0.959 and 0.965), respectively. The correlation matrix also showed strong correlation between the ABTS and FRAP values (r = 0.992). Furthermore, a multivariate analysis (PCA), which was extracted from the data of Table 1, was carried out. As shown in Figure 2, PCA explained 94.44% of total variation, in which PC1 accounted for 84.73% of the variance and PC2 accounted for 9.71%. The FRAP, ABTS, and DPPH assays with EAF and RWF are placed at the upper right quadrants, and TPC and TFC concentrations with BF and BF-C to E are located at the lower right quadrants in the PC1 positive scores, respectively. Meanwhile, the DF and BF-A to B with low TPC and TFC concentrations are located along the axis of PC1 negative scores. These findings showed that the TPC and/or TFC concentrations are closely associated with antioxidant capacity, and the greater the TPC or TFC concentrations in the fractions and subfractions, the higher their antioxidant capacity values. These results revealed that the high phenolic and flavonoid contents in different fractions and subfractions from *A. fragrans* leaves might contribute to antioxidant activity.

### 2.7. Inhibitory Effect on Intracellular ROS Generation

Oxidative stress is a kind of imbalance between oxidation and antioxidation in the body [28]. Excessive accumulation of ROS can lead to oxidative stress that may cause damage to cells and tissues such as lipids, membranes, and DNA [29]. Antioxidants including polyphenols and flavonoids can help the body reduce these damages caused by ROS. Generally, H_2_O_2_ is widely used to induce intracellular ROS disordered production and impair the antioxidant defense of cells [30]. In our present study, H_2_O_2_ was chosen to induce the abnormal accumulation of intracellular ROS and impair the antioxidant defense of cells. To evaluate the inhibitory effect on intracellular ROS generation in H_2_O_2_-induced HepG2 cells, the levels of intracellular ROS were tested by flow cytometry. Briefly, HepG2 cells were cultured in a 6-well plate (1 × 10^5^ cells/well). After incubation for 24 h of 50 μg/mL of different extracts or 8 μg/mL of Vc (positive group), all the groups except for the control group were induced by H_2_O_2_ for 6 h. The intracellular ROS levels of each compound were tested (Figure 3). The ROS generation ratio significantly increased to 215.64 ± 9.80% in H_2_O_2_-treated group compared with the control group (100%). Compared with the H_2_O_2_-treated group, compounds **2**, **5**, and **6** remarkably suppressed intracellular ROS production (*p* < 0.05), and their inhibitory effect was equal to the Vc group (Figure 3). Many phenolics have been proven to have protective effect against intracellular ROS by H_2_O_2_ induction [31]. Additionally, compound **6** displayed the strongest suppressive effect on intracellular ROS production, which suggests that flavonoid compounds play a crucial role in inhibiting intracellular ROS production.

### 2.8. Cytoprotective Effect Against H_2_O_2_-Induced Cell Apoptosis

Apoptosis is a basic biological phenomenon of cells, which plays an important role in the regulatory mechanism of cells’ proliferation, growth, and mutation, and the stability of the internal environment. Apoptosis, different from necrosis, is a special type of cell death. The disorder of apoptotic process has bad effects on the body and causes many diseases [32]. H_2_O_2_, as an important signaling molecule, regulates the process of cell proliferation, growth, and apoptosis [5]. The present study measured the apoptosis of H_2_O_2_-induced HepG2 cells and evaluated the cytoprotective effects of compounds **2**, **5**, and **6**. After treating HepG2 cells with 1.0 mM H_2_O_2_, the apoptosis ratio remarkably augmented (57.20 ± 1.97%), compared with that of the control group (9.10 ± 0.62%, *p* < 0.05) (Figure 4). The ratios of apoptotic cells in the treated groups of compounds **2**, **5**, and **6** significantly decreased compared with the H_2_O_2_-treated group (model group, *p* < 0.05) (Figure 4). Moreover, compound **6** had significantly efficiency on protecting HepG-2 cells from H_2_O_2_ toxicity, and the cell apoptosis ratio of **6** (10.56 ± 1.15%) was lower than that of the Vc group (positive control), which was equal to that of the control group (Figure 4). Meanwhile, compounds **2** and **5** showed moderate cytoprotective effect with cell apoptosis ratios of **2** (26.76 ± 2.60%) and **5** (27.64 ± 0.83%). The differences of antioxidant capacity may be attributed to the number of phenolic hydroxyl moieties and the link positions.

## 3. Materials and Methods

### 3.1. Chemicals and Reagents

Methanol, acetonitrile, and formic acid for high-performance liquid chromatography (HPLC) were of HPLC grade and purchased from Merck (Darmstadt, Germany). Solvents for sample extraction including ethanol, dichloromethane, and *n*-butanol were of analytical grade. Deionized water was purified using a Milli-Q ultrapure water system (Millipore, Bedford, Massachusetts, MA, USA) and employed in all the experiments. Phenolic standard compounds of gallic acid, rutin, Trolox, and vitamin C were purchased from Chengdu Must Bio-Technology Co., Ltd. (Chengdu, China). Methylthiazol-2-yl-2,5-diphenyl tetrazolium bromide (MTT), Folin–Ciocalteu reagent, 2,2ʹ-azino-bis(3-ethylbenzo-thiazoline-6-sulfonic acid) (ABTS), 2,2-diphenyl-1-picrylhydrazyl radical (DPPH), 1,3,5-tri(2-pyridyl)-2,4,6-triazine (TPTZ), 2ʹ,7ʹ-dichlorofluorescin diacetate (DCFH-DA), and FeSO_4_·7H_2_O were purchased from Sigma–Aldrich (Shanghai, China). The NMR spectra were obtained using Bruker AV-400, and/or DRX-500 spectrometers. ESIMS spectra were recorded on An Agilent 1290 UPLC/6540 Q-TOF spectrometer.

### 3.2. Sample Preparation

*Anneslea fragrans* Wall. leaves were collected from the Lincang city of China in July 2020. The leaves were dried in a shade room until constant weight and then were powdered with an electric grinder. The extraction and fractionation were performed as our previously reported method with slight modification [33]. The powdered sample (100 g) was mixed with 1000 mL of 80% aqueous ethanol solvent and ultrasonicated in an ultrasonic cleaning bath at 200 W for three times (0.5 h per time). Then, the sonicated slurry was collected and centrifuged at 1500× *g* for 10 min by Eppendorf centrifuge (TGL-20B, Shanghai Anting Scientific Instrument Factory, Shanghai, China). The combined supernatant was concentrated at 50 °C by a rotary evaporator (Hei-VAP, Heidolph, Germany) and further dried by a vacuum drying lyophilizer (Alpha 1-2 LD plus, Christ, Germany). The crude ethanol extract (CE, 30 g) was re-suspended with water and sequentially partitioned with dichloromethane, ethyl acetate, and *n*-butanol solvents three times. After concentration and lyophilization, the dichloromethane fraction (DF), ethyl acetate fraction (EAF), *n*-butanol fraction (BF), and residual water fraction (RWF) weighing 3.2 g, 6.3 g, 7.2 g, and 8.2 g were obtained, respectively. According to the antioxidant activities of different fractions, the BF was chromatographed on glass columns (30 mm × 400 mm) wet-packed with 20 g (dry resin) of the selected hydrated resin D101. The bed volume (BV) of the resin was about 40 mL. After reaching the adsorptive saturation, the column was first washed by distilled water with 4 × BV and then eluted by ethanol–water (0:100, 20:80, 50:50, 80:20, 100:0, *v*/*v*, each 4 × BV), to yield five subfractions (BF-A to E). Each part of the desorption solutions was concentrated to dryness under vacuum. The CE, four fractions (DF, EAF, BF, and RWF) and five subfractions (BF-A to E) were stored in a refrigerator (−20 °C) for further experimentation.

### 3.3. Bio-Guided Isolation of Active Constituents

Under the guidance of antioxidant assays and HPLC analysis, the antioxidative fraction was further chromatographed for the isolation of pure compounds. In brief, the BF was subjected to a hydrated resin D101 column to yield five subfractions (BF-A to E). The BF-C (1.5 g) was subjected to a silica gel column, eluting with DCM/MeOH (15:1), and then was separated using preparative TLC (DCM/MeOH, 10:1) to obtain compounds **6** (118 mg) and **7** (10 mg). BF-D (1.1 g) was subjected to silica gel column (CHCl_3_/MeOH 10:1, 5:1) to give compounds **1** (129 mg) and **4** (15 mg). BF-E (1.2 g) was subjected to silica gel column (CHCl_3_/MeOH, 30:1, 10:1, 5:1) to yield compounds **1** (216 mg), **2** (135 mg), and a mixture. The later was purified by silica gel column (CHCl_3_/MeOH, 12:1) to afford compounds **3** (19 mg) and **5** (15 mg) (Figure 5).

### 3.4. Structure Elucidation of Compounds ***1***–***7***

According to the antioxidant activities of different fractions, three subfractions, BF-C to E were submitted to column chromatography. In this way, the bioactivity-guided fractionation of BF-C to E led to the isolation of seven individual phenolic compounds. Their structures were identified as confusoside (**1**) [34], vacciniifolin (**2**) [34], 1-[4-(β-d-glucopyranosyloxy)-2-hydroxyphenyl]-3-(4-hydroxy-3-methoxyphenyl)-1-propanone (**3**) [35], (*S*)-naringenin-7-*O*-*β*-d-glucopyranoside (**4**) [22], 2′,3,4,4′-tetrahydroxydihydrochalcone (**5**) [26], (epi)-catechin (**6**) [26], and cornoside (**7**) [36] (Figure 1B) by the analysis of 1D-NMR and ESI-MS data and comparison with the previously reported compounds in the literature. Among them, compounds **3**, **4**, and **7** were isolated from this plant for the first time.

Confusoside (**1**). The molecular formula was assigned as C_21_H_24_O_9_, and the separation gave 129 mg of pale-yellow needles. ^1^H NMR (500 MHz, DMSO-*d*_6_) and ESI-MS (*m*/*z* 421 [M + H]^+^) data were identical to the previously reported compound in the literature [34]. The identification was further supported by ^13^C NMR (125 MHz, DMSO-*d*_6_): δ 204.4 (s, C=O), 163.5 (s, C-2′), 163.3 (s, C-4′), 155.5 (s, C-4), 132.6 (d, C-6′), 130.9 (s, C-1), 129.2 (d, C-2, 6), 115.0 (d, C-3/C-5), 114.4 (s, C-1′), 108.3 (d, C-5′), 103.4 (d, C-1″), 99.6 (d, C-3′), 77.1 (d, C-5″), 76.4 (d, C-3″), 73.1 (d, C-2″), 69.5 (d, C-4″), 60.5 (t, C-6″), 39.8 (t, C-*α*), 28.9 (t, C-*β*).

Vacciniifolin (**2**) was obtained as yellow amorphous powder (153 mg) and assigned a molecular formula of C_21_H_24_O_10_. ^1^H NMR (500 MHz, DMSO-*d*_6_) and ESI-MS (*m*/*z* 437 [M+H]^+^) data were the same as the previous reported data [34]. The identification was further supported by ^13^C NMR (125 MHz, DMSO-*d*_6_): δ 204.4 (s, C=O), 163.5 (s, C-2′), 163.3 (s, C-4′), 145.0 (s, C-3), 143.3 (s, C-4), 132.6 (d, C-6′), 131.7 (s, C-1), 118.9 (d, C-6), 115.8 (d, C-2), 115.4 (d, C-5), 114.4 (s, C-1′), 108.3 (d, C-5′), 103.4 (d, C-1″), 99.6 (d, C-3′), 77.1 (d, C-5″), 76.4 (d, C-3″), 73.1 (d, C-2″), 69.5 (d, C-4″), 60.5 (t, C-6″), 39.4 (t, C-*α*), 29.1 (t, C-*β*).

1-[4-(β-D-Glucopyranosyloxy)-2-hydroxyphenyl]-3-(4-hydroxy-3-methoxyphenyl)-1-propanone (**3**). The compound was obtained as colorless needles (19 mg), and the molecular formula was assigned as C_22_H_26_O_10_. ^1^H NMR (400 MHz, DMSO-*d*_6_) and ESI-MS (*m*/*z* 451 [M + H]^+^) data agreed with the literature [35]. The identification was further supported by ^13^C NMR (100 MHz, DMSO-*d*_6_): δ 204.5 (s, C=O), 163.5 (s, C-2′), 163.3 (s, C-4′), 147.4 (s, C-3), 144.6 (s, C-4), 132.6 (d, C-6′), 131.6 (s, C-1), 120.4 (d, C-6), 115.2 (d, C-5), 114.5 (s, C-1′), 112.6 (d, C-2), 108.3 (d, C-5′), 103.3 (d, C-1″), 99.5 (d, C-3′), 77.0 (d, C-5″), 76.3 (d, C-3″), 73.0 (d, C-2″), 69.5 (d, C-4″), 60.5 (t, C-6″), 55.5 (q, C-OCH_3_), 39.5 (t, C-*α*), 29.4 (t, C-*β*).

(*S*)-Naringenin-7-*O*-*β*-d-glucopyranoside (**4**) had the molecular formula of C_21_H_22_O_10_, which was obtained as 15 mg of white powder. ^1^H NMR (500 MHz, DMSO-*d*_6_) and ESI-MS (*m*/*z* 435 [M + H]^+^) data were in agreement with the previous work [22]. The identification was further supported by ^13^C NMR (125 MHz, DMSO-*d*_6_): δ 197.2 (s, C-4), 165.3 (s, C-7), 165.2 (s, C-5), 162.9 (s, C-9), 157.9 (s, C-4′), 128.6 (s, C-1′), 128.4 (d, C-2′, 6′), 115.2 (d, C-3′/C-5′), 103.2 (s, C-10), 99.6 (d, C-1″), 96.5 (d, C-6), 95.4 (d, C-8), 78.7 (d, C-2), 77.0 (d, C-5″), 76.3 (d, C-3″), 73.0 (d, C-2″), 69.5 (d, C-4″), 60.5 (t, C-6″), 42.0 (t, C-3).

2′,3,4,4′-Tetrahydroxydihydrochalcone (**5**) possessed the molecular formula as C_15_H_14_O_5_ and separated (15 mg) as white powder. ^1^H NMR (400 MHz, DMSO-*d*_6_) and ESI-MS (*m*/*z* 275 [M + H]^+^) data were consistent with that reported in the literature [26]. The identification was further supported by ^13^C NMR (125 MHz, DMSO-*d*_6_): δ 203.9 (s, C=O), 164.7 (s, C-2′), 164.2 (s, C-4′), 144.9 (s, C-3), 143.3 (s, C-4), 133.0 (d, C-6′), 131.8 (s, C-1), 118.9 (d, C-6), 115.7 (d, C-2), 115.4 (d, C-5), 112.5 (s, C-1′), 108.2 (d, C-5′), 102.4 (d, C-3′), 39.5 (t, C-*α*), 29.2 (t, C-*β*).

(Epi)-catechin (**6**) was isolated as white powder (118 mg) and established the molecular formula as C_11_H_6_O_4_. ^1^H NMR (400 MHz, DMSO-*d*_6_) and ESI-MS (*m*/*z* 291 [M + H]^+^) data agreed well with that reported in the literature [26]. The identification was further supported by ^13^C NMR (125 MHz, DMSO-*d*_6_): δ 156.4 (s, C-7), 156.1 (s, C-5), 155.3 (s, C-9), 144.8 (s, C-3′), 144.8 (s, C-4′), 130.5 (s, C-1′), 118.4 (d, C-6′), 115.0 (d, C-5′), 114.4 (d, C-2′), 99.0 (s, C-10), 95.1 (d, C-6), 93.8 (d, C-8), 80.9 (d, C-2), 66.2 (d, C-3), 27.8 (t, C-4).

Cornoside (**7**) was obtained as amorphous solid (10 mg) and determined the molecular formula as C_14_H_20_O_8_. ^1^H NMR (500 MHz, DMSO-*d*_6_) and ESI-MS (*m*/*z* 317 [M + H]^+^) data corresponded with the published data [36]. The identification was further supported by ^13^C NMR (100 MHz, DMSO-*d*_6_): δ 185.3 (s, C-4), 153.3 (d, C-2), 153.2 (d, C-6), 126.4 (d, C-3), 126.4 (d, C-5), 102.8 (d, C-1′), 76.8 (d, C-5′), 76.6 (d, C-3′), 73.3 (d, C-2′), 70.0 (d, C-4′), 67.3 (s, C-1), 63.8 (t, C-8), 61.0 (t, C-6′), 39.7 (t, C-7).

### 3.5. Determination of Total Phenolic (TPC) and Total Flavonoid Contents (TFC)

The TPC and TFC of four fractions (DF, EAF, BF, and RWF) and five subfractions (BF-A to E) were measured according to our previously reported method [37]. For TFC, 1.0 mL of each sample (with the concentration at 1.0 mg/mL) dissolved in methanol was mixed with 0.5 mL of Folin–Ciocalteu reagent in a centrifuge tube and incubated for 1 min. Then, 20% Na_2_CO_3_ solution (*m/v*) (1.5 mL) and deionized water (7.0 mL) were added to the tube and kept at 70 °C in a water bath for 10 min. After being cooled to room temperature, 200 μL of the solution was transferred to a 96-well microplate and the absorbance was determined at 765 nm by a SpectraMax M5 microplate reader (Molecular Devices, Sunnyvale, CA, USA).

For TFC, 1.2 mL of sample solutions (with the concentration at 1.0 mg/mL) were mixed with 0.3 mL of NaNO_2_ (5% *m/v*) and 3.8 mL of 70% aqueous ethanol and incubated for 8 min. Subsequently, 0.3 mL 10% aqueous Al(NO_3_)_3_, 4 mL 4% aqueous NaOH, and 0.4 mL 70% aqueous ethanol were added to the mixture and allowed to react at room temperature for 30 min. Then, 200 μL of the solution was transferred to a 96-well microplate, for which the absorbance was measured at 510 nm by using a microplate reader. The TFC and TPC was expressed as milligrams of gallic acid equivalents (mg GAE/g extract) and rutin equivalents per gram of extract (mg RE/g extract).

### 3.6. Determination of Antioxidant Activity

The antioxidant activity of four fractions (DF, EAF, BF, and RWF) and five subfractions (BF-A to E) were evaluated in a combination of DPPH and ABTS radical scavenging assays and FRAP assay based on the method described in our previous study [33]. For DPPH assay, 50 μL of the sample solution (50, 100, 200 μg/mL) was mixed with 0.2 mL DPPH solution (0.1 mmol/L) in a 96-well plate and allowed to incubate for 30 min. The absorbance was measured at 517 nm with SpectraMax M5 microplate reader (Molecular Devices, Sunnyvale, CA, USA). For ABTS, 25 μL of the sample solution (50, 100, 200 μg/mL) were added to 0.2 mL ABTS solution (7 mmol/L). The mixture was kept in the dark for 6 min, and then, the absorbance was recorded at 734 nm. For FRAP, 20 μL sample solution (50, 100, 200 μg/mL) was mixed with 0.18 mL of FRAP reagent (7 mmol/L). After incubating for 10 min in the dark at 37 °C, the absorbance was determined at 593 nm. All the tests were performed in triplicate. The results of DPPH, ABTS, and FRAP values were expressed as μmol Trolox equivalents per gram of extract (µmol TE/g extract).

### 3.7. HPLC Analysis

The BF and compounds **1**–**7** were analyzed on an Agilent 1260 HPLC system coupled with a diode array detector. Before analysis, the freshly prepared sample solution was filtered through a 0.45 μm nylon membrane. The separation was performed using a Reprosil-Pur Basic C18 column (5 μm, 4.6 × 250 mm, Germany) maintained at 35 °C. The injection volume was 5.0 μL, the flow rate was 1.0 mL/min, and the detection wavelength was set at 280 nm. The mobile phases were acidified water with 0.1% formic acid (phase A) and acetonitrile (phase B), the linear gradient elution was performed as follows: 0–3 min, 20% B; 3–10 min, 40% B; 10–15 min, 60% B; 15–20 min, 100% B.

### 3.8. Cell Culture and Cell Viability

Human liver cancer HepG2 cells were purchased from Kunming Cell Bank (Kunming, China). HepG2 cells were grown in DMEM supplemented with 1% penicillin–streptomycin and 10% fetal bovine serum in an atmosphere of 5% CO_2_/95% air at 37 °C. When the cells were incubated to an appropriate density (approximately 80%), they were treated with positive control (Vc) and the isolated compounds for further experiments.

Cell viability was determined by MTT assay for evaluating the cytotoxicity of each sample [29]. The cells at a density of 1 × 10^4^ cells per well were seeded in a 96-well plate and allowed to incubate for 24 h. Each compound (prepared as four doses from 50 to 200 μg/mL) was added to each well for 20 h. Then, the cells were treated using MTT solution with a final concentration for 4 h. The medium with MTT was removed, and 200 μL of the DMSO was added to dissolve the formazan. The absorbance was recorded at 570 nm by a microplate reader. The results demonstrated that each compound was nontoxic to HepG2 cells at the tested concentrations.

### 3.9. Inhibition of ROS Generation in H_2_O_2_-Induced HepG2 Cells

H_2_O_2_-induced HepG2 cells were employed to determine the inhibitory effect on ROS production [3]. HepG2 cells (1.0 × 10^5^ cells per well) were seeded in a 6-well plate and co-cultured with isolated compounds with 50 μg/mL and Vc (8 μg/mL). After incubation for 20 h, the medium was removed, and 2 μL of H_2_O_2_ (0.5 mM) was added to each well for another 6 h. At the end of experiment, the cells were labeled with 2 μL DCFH-DA (10 mM) in the dark at 37 °C for 20 min. The absorbance was recorded by flow cytometry (Guava easyCyte 6-2L, Millipore, Billerica, Massachusetts, MA, USA).

### 3.10. Determination of Cell Apoptosis

The protective effect of each compound on H_2_O_2_-induced apoptosis of HepG2 cells was determined using a human annexin VFITC/PI apoptosis kit [38]. HepG2 cells were pre-treated with or without isolated compounds for 48 h. After incubation, 100 μL of the binding buffer was added to the cells, and the cells reacted in the dark with 10 μL annexin V-FITC for 5 min at room temperature and with 10 μL propidium iodide (PI) in an ice bath for 5 min, successively. Cell apoptosis was immediately analyzed using flow cytometry.

### 3.11. Statistical Analysis

All the experiments were performed in triplicate. All the values are expressed as mean ± standard deviation (SD). The differences within and between the groups were analyzed using one-way analysis of variance (ANOVA) followed by Tukey’s test. Difference was considered statistically significant at *p* < 0.05. All analyses were performed using Origin 2019b software (OriginLab, Northampton, MA, USA).

## 4. Conclusions

In this study, different fractions from *A. fragrans* leaves were fractionated, and their TPC and TFC were analyzed. Under antioxidant activity guided isolation, compounds **1**−**7**, including four flavonoid glycosides (**1**–**4**) and two flavonoids (**5** and **6**), were isolated and identified from *A. fragrans* leaves, which suggested that this species is rich in flavonoid compounds. Compounds **2**, **5**, and **6** showed significant antioxidant activity in DPPH, ABTS radical scavenging, and FRAP assays. Furthermore, they visibly prevented the oxidative stress damage through a decrease in ROS content and cell apoptosis in H_2_O_2_-induced HepG2 cells. According to these results, polyphenol compounds, especially flavonoids, have considerable antioxidant capacity because of their phenolic hydroxyl groups. Furthermore, compound **2**, possessing the glycoside moiety and three phenolic hydroxyl groups, was the main antioxidant component with the highest content from *A. fragrans* leaves. Compound **6** displayed the best antioxidant activity, which may be a major contribution to the activity of *A. fragrans.* Furthermore, the extracts of *A. fragrans* could be served as a feasible natural source of antioxidants in promising health beverages. The study on the compounds from *A. fragrans* leaves suggest that these could be served as antioxidant healthy tea for treating oxidative stress-induced cell damage and could serve as nutritional supplements applied in the food and health industry.

## Figures and Tables

**Figure 1 molecules-26-03690-f001:**
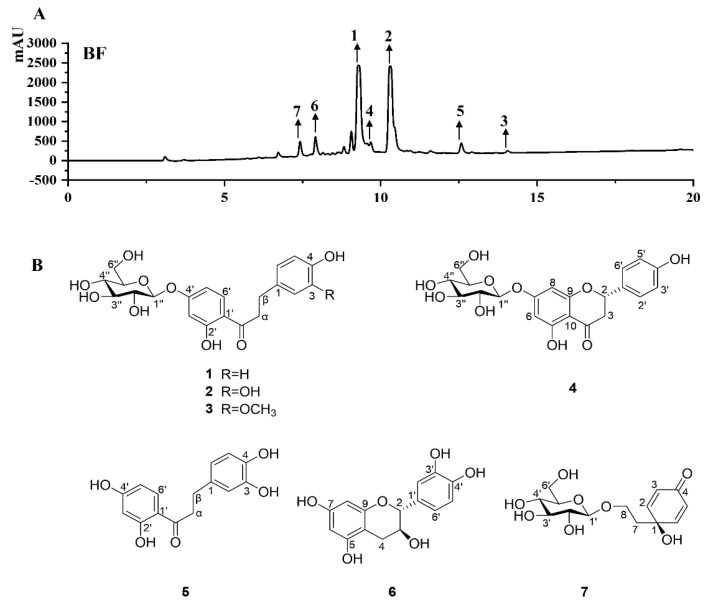
(**A**). HPLC chromatogram of BF from *Anneslea fragrans.* (**B**). Chemical structures of the compounds **1**–**7**.

**Figure 2 molecules-26-03690-f002:**
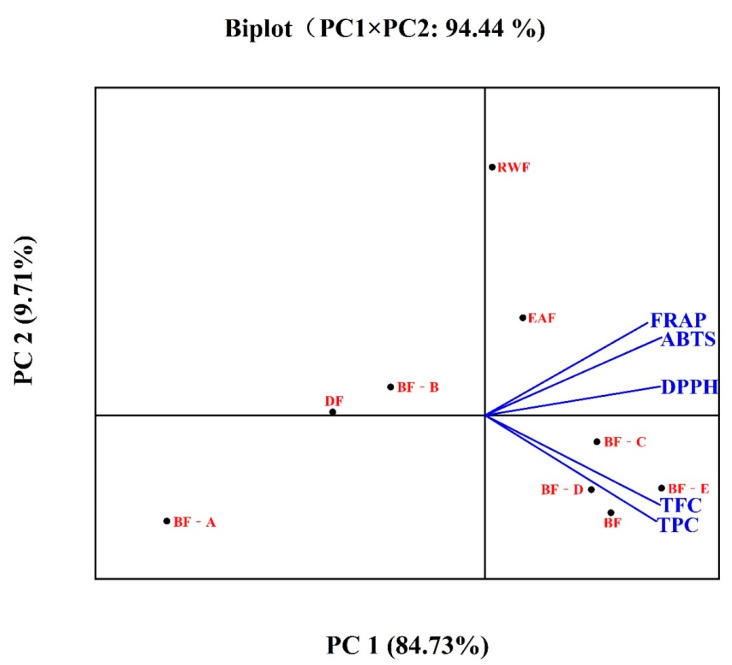
PCA analysis on total phenolics, flavonoids, and antioxidant activity. Means with different letters indicate significant differences (*p* < 0.05). DF, EAF, BF, RWF mean dichloromethane fraction, ethyl acetate fraction, *n*-butanol fraction, and residual water fraction. BF-A to E means BF was subjected to D101 column chromatography to yield five subfractions.

**Figure 3 molecules-26-03690-f003:**
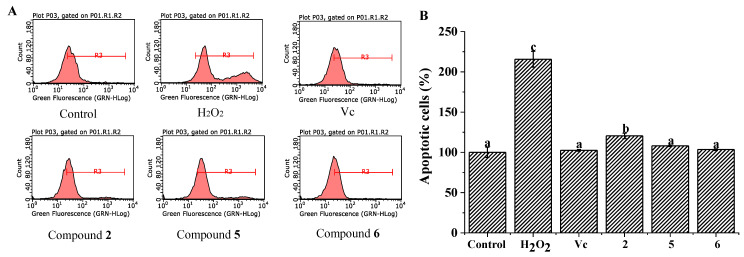
The inhibitory effects of compounds **2**, **5**, and **6** on intracellular ROS in H_2_O_2_-induced HepG2 cells. (**A**) Flow cytometry analysis; (**B**) the ROS intensity of different groups. All the values are presented as mean ± SD (*n* = 3). Means (bar values) with different letters indicate significant differences (*p* < 0.05).

**Figure 4 molecules-26-03690-f004:**
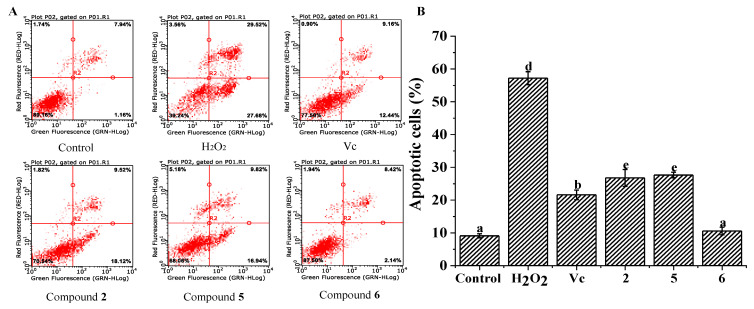
Cytoprotective effect of the compounds **2**, **5**, and **6** on apoptosis in H_2_O_2_-induced HepG-2 cells. (**A**) Flow cytometry analysis; (**B**) the apoptotic cell percentage of different groups. All the values are presented as mean ± SD (*n* = 3). Means (bar value) with different letters are significantly different (*p* < 0.05).

**Figure 5 molecules-26-03690-f005:**
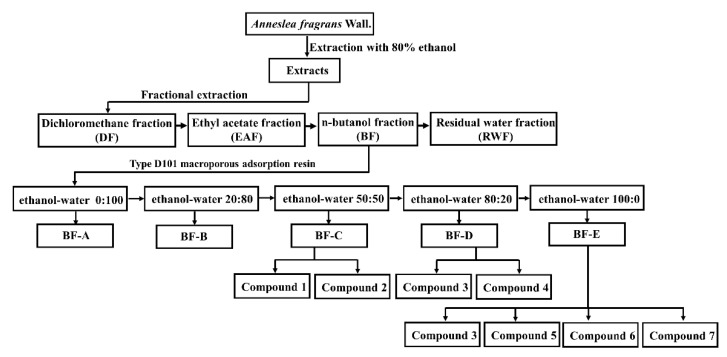
Extraction and isolation procedure of the compounds from 80% aqueous ethanol extract from *Anneslea fragrans* leaves.

**Table 1 molecules-26-03690-t001:** TPC, TFC, and antioxidant capacity values of four fractions and five subfractions from *Anneslea fragrans* Wall ^1^.

Samples	TPC	TFC	FRAP ^2^	ABTS ^3^	DPPH ^4^
Vc ^5^	-	-	3126.08 ± 197.63 ^h^	2932.91 ± 93.63 ^g^	1873.56 ± 121.68 ^a^
DF	66.52 ± 0.57 ^b^	47.94 ± 2.84 ^b^	180.05 ± 9.46 ^b^	139.23 ± 8.62 ^a^	779.94 ± 84.76 ^b^
EAF	123.49 ± 1.14 ^d^	116.12 ± 2.89 ^d^	975.24 ± 16.73 ^e^	796.92 ± 19.26 ^c^	684.31 ± 31.26 ^b^
BF	238.12 ± 12.05 ^g^	165.19 ± 5.21 ^f^	1822.96 ± 29.24 ^f^	1808.46 ± 96.52 ^d^	951.42 ± 87.75 ^b^
RWF	81.29 ± 1.48 ^c^	66.92 ± 1.38 ^c^	592.65 ± 11.55 ^d^	135.38 ± 6.75 ^a^	613.38 ± 45.35 ^b^
BF-A	22.22 ± 1.63 ^a^	25.46 ± 2.89 ^a^	71.27 ± 5.42 ^a^	130.01 ± 4.16 ^a^	128.60 ± 6.31 ^a^
BF-B	76.37 ± 0.94 ^c^	63.63 ± 3.82 ^c^	427.61 ± 13.78 ^c^	246.92 ± 7.03 ^b^	615.31 ± 38.21 ^b^
BF-C	210.69 ± 6.31 ^e^	151.01 ± 4.33 ^e^	1777.94 ± 72.93 ^f^	2020.87 ± 97.29 ^e^	1249.87 ± 81.59 ^c^
BF-D	219.84 ± 4.01 ^f^	157.01 ± 2.50 ^e^	1839.38± 12.87 ^f^	1881.53 ± 28.57 ^d^	1431.35 ± 98.32 ^cd^
BF-E	262.03 ± 1.72 ^h^	180.52 ± 6.30 ^g^	2096.77 ± 49.13 ^g^	2162.31 ± 64.49 ^f^	1660.42 ± 119.72 ^de^

^1^ TPC: total phenolic content expressed as mg GAE/g extract, TFC: total flavonoid content expressed as mg RE/g extract. All the values are expressed as mean ± SD (*n* = 3). Data are obtained from three replicates and presented as mean ± SD; different numbers in the same column with different letters as superscript are significantly different (*p* < 0.05). DF, EAF, BF, and RWF mean dichloromethane fraction, ethyl acetate fraction, *n*-butanol fraction, and residual water fraction. BF-A to E mean that BF was subjected to D101 column chromatography to yield five subfractions. ^2^ FRAP: μmol Trolox equivalents (TE)/g extract. ^3^ ABTS: μmol Trolox equivalents (TE)/g extract. ^4^ DPPH: μmol Trolox equivalents (TE)/g extract. ^5^ Vc means the standard of vitamin C.

**Table 2 molecules-26-03690-t002:** Antioxidant capacity values of seven compounds from *Anneslea fragrans* Wall ^1^.

Compound	FRAP ^2^	ABTS ^3^	DPPH ^4^
Vc ^5^	3126.08 ± 197.63 ^e^	2932.91 ± 93.63 ^e^	1873.56 ± 121.68 ^c^
**1**	714.17 ± 64.63 ^b^	174.81 ± 14.38 ^a^	133.33 ± 4.98 ^a^
**2**	3065.37 ± 283.87 ^e^	1312.41 ± 108.63 ^c^	1888.31 ± 162.94 ^c^
**3**	562.09 ± 63.98 ^a^	126.37 ± 9.32 ^a^	128.33 ± 9.93 ^a^
**4**	2806.15 ±171.92 ^d^	1021.92 ± 51.82 ^b^	1418.31 ± 84.62 ^b^
**5**	2448.36 ± 150.38 ^c^	1132.16 ± 93.85 ^b^	1533.31 ± 114.96 ^b^
**6**	5027.43 ± 620.75 ^f^	1580.37 ± 89.32 ^d^	1953.31 ± 109.93 ^c^
**7**	613.94 ± 87.29 ^a^	54.18 ± 4.38 ^a^	108.33 ± 8.71 ^a^

^1^ TPC: total phenolic content expressed as mg GAE/g extract, TFC: total flavonoid content expressed as mg RE/g extract. Data are obtained from three replicates and presented as mean ± SD; different numbers in the same column with different letters as superscript are significantly different (*p* < 0.05). ^2^ FRAP: μmol Trolox equivalents (TE)/g extract. ^3^ ABTS: μmol Trolox equivalents (TE)/g extract. ^4^ DPPH: μmol Trolox equivalents (TE)/g extract. ^5^ Vc means the standard of vitamin C.

## Data Availability

Data is contained within the article.

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
