# Peer review of "Activity Guided Isolation of Phenolic Compositions from Anneslea fragrans Wall. and Their Cytoprotective Effect against Hydrogen Peroxide Induced Oxidative Stress in HepG2 Cells"

_molecules, 2021, doi:10.3390/molecules26123690_

Round 1
Reviewer 1 Report
Although potentially interesting the paper entitled "Activity guided isolation of phenolic compositions from Anneslea fragrans Wall. and their cytoprotective effect against hydrogen peroxide induced oxidative stress in HepG2 cells" has 2 major drawbacks which are its structure and the English language. I have not been able to follow the paper in any moment. In its present form it is completely unintelligible. Despite there are some interesting results, the paper should be fully re-structured and deeply re-written.
Author Response
Response: Thank you so much for your careful comments on our paper. We have carefully read the paper and your detailed comments. We have modified it one by one according to your suggestion. The purpose of this research was to isolate and identify antioxidant compounds from A. fragrans leaves which are responsible for its traditional use for the treatment of liver diseases. Firstly, four fractions and five subfractions were evaluated for their total phenolic (TPC) and flavonoid contents (TFC), and antioxidant activities (DPPH, ABTS and FRAP assays). Secondly, Bio-guided fractionation of n-butanol fraction (BF) allowed the purification of pure compounds. Furthermore, the cytoprotective effect of the isolates was performed on hydrogen peroxide (H2O2)-induced oxidative stress in human liver cancer HepG2 cells.
In the Revised Manuscript, all changes have been marked in red. We reworked and polished the language and changed the structure. In addition, we have asked Afsar Khan (one co-author), whose mother language is English, polished the writing of our revised manuscript.
Thank you very much!
Reviewer 2 Report
The manuscript is dealing with isolation and characterization of phenolic compositions from Anneslea fragrans Wall. and futher studies for their cytoprotective effect against hydrogen peroxide induced oxidative stress in HepG2 cells.
Nevertheless, the manuscript needs to be improved in order to provide enough information to justify its importance and novelty.
Specific comments
- line 63 please describe what means DF, EAF, BF and RWF?
- Table 1 - please explain the differences between TPC and TFC
- Table 1 TPC and TFC were expresses as???
- Table 1 please express FRAP, ABTS and DPPH as per g DW in order to be able to compare them, or do it for all as IC50
- line 289 - lyophilizer
- why did you choose combination of dichloromethane, ethyl acetate, and n-butanol solvents (1:1, v/v). The proportion is given anly fo 2 solvents.
- all yourb results in order to compare them should be express as per g DW.
- Please use the section background to explain the importance and novelty of this research and its objective.
- Please conclude according to the main objectives of the research.
Author Response
Response: Thank you for your suggestion, which are great and helpful to us. We have carefully read the paper and your detailed comments. In the Revised Manuscript, all changes have been marked in red. We would like to express our great appreciation to you for comments on our paper. In addition to the grammatical errors, I would like to reply to the suggestions you made in the attached manuscript as follows:
- Line 63 please describe what means DF, EAF, BF and RWF?
Response: Thank you for your suggestion. The sentence was rewritten as “The four fractions, dichloromethane fraction (DF), ethyl acetate fraction (EAF), n-butanol fraction (BF) and residual water fraction (RWF) of ethanol-aqueous extract were assessed their total phenolic (TPC) and total flavonoid contents (TFC), and their antioxidant capacities.”
- Table 1 - please explain the differences between TPC and TFC.
Response: Thanks for your suggestion. The “TPC” means total phenolic content and the “TFC” means total flavonoid content. Total phenolics are secondary metabolites ubiquitously distributed in many higher plants. As regard to chemical structure, they comprise a wide variety of molecules with polyphenol structure and are generally divided into flavonoids and nonflavonoids. The reaction principle of total phenolics is that phenolic compounds can reduce tungstomolybdic acid under alkaline conditions to produce blue compounds. The depth of color is positively correlated with the content of phenolic compounds, and the maximum absorption occurs at the wavelength of about 760 nm. Total flavonoids share a common carbon skeleton of diphenyl propanes, two benzene rings (ring A and B) joined by a linear three-carbon chain. The central three-carbon chain forms a closed pyran ring (ring C) with A benzene ring. Depending on the oxidation state of the central pyran ring, flavonoids can themselves be subdivided into many subclasses: flavonols, flavones, flavanones, anthocyanidins, flavanols, and also isoflavones. Total flavonoids are determined by the principle of 3-hydroxyl, 4-hydroxyl or 5-hydroxyl, 4-carbonyl or o-diphenolic hydroxyl, complex reaction with aluminum salts, under alkaline conditions to form a red complex. The main groups of nonflavonoids are first phenolic acids, second stilbenes, and third lignans. In addition to this diversity, total phenolic are present in plant tissues mainly as glycosides and/or associated with various organic acids and/or as complex polymerized molecules with high molecular weights, such as tannins.
- Table 1 TPC and TFC were expresses as???
Response: Thanks for your suggestion. The TPC and TFC were expresses as “total phenolic content” and “total flavonoid content”.
- Table 1 please express FRAP, ABTS and DPPH as per g DW in order to be able to compare them, or do it for all as IC50.
Response: Thanks for your suggestion. The antioxidant capacity values were expressed using FRAP, ABTS and DPPH as per g DW in Table 1.
- Line 289 - lyophilizer
Response: Thanks for the pointing out the error. The word is revised to “lyophilizer”.
- why did you choose combination of dichloromethane, ethyl acetate, and n-butanol solvents (1:1, v/v). The proportion is given anly fo 2 solvents.
Response: Thanks for your comment. The ethanol extract (CE) from A. fragrans leaves was successively partitioned with dichloromethane, ethyl acetate, and n-butanol solvents for three times. The water layer and the organic solvent layer were given each time. And there are not two solvents used for extraction.
- All your results in order to compare them should be express as per g DW.
Response: Thanks for your suggestion. Our results have been expressed in terms of g DW.
- Please use the section background to explain the importance and novelty of this research and its objective.
Response: Thanks for your suggestion. The leaves of Anneslea fragrans Wall. are also used as a folk medicine to treat fever, liver protection, invigorating stomach and intestines in China and Cambodia, which had been recorded in “Yunnan Simao Chinese Herbal Medicine”. In addition, the leaves have also been processed as a tea beverage, known as “Pangpo Tea”. Recently, many studies have evidenced that the intake of diet enriched with antioxidants possess a series of beneficial effects owing to their scavenging ability on excessive ROS. Polyphenols are well known for their protective effect by scavenging ROS. Therefore, searching for effective antioxidants is an urgent need to promote human health. Although in previous reports, the extract of A. fragrans has shown antioxidant and antimalarial activities. However, to date, its antioxidant phenolic compounds from A. fragrans have not been reported. Thus, the purpose of this research was to isolate and identify antioxidant compounds from A. fragrans leaves which are responsible for its traditional use for the treatment of liver diseases. Bio-guided fractionation of n-butanol fraction (BF) allowed the purification of pure compounds. Furthermore, the cytoprotective effect of the isolates was performed on hydrogen peroxide (H2O2)-induced oxidative stress in human liver cancer HepG2 cells. Thus, this research afforded a valuable antioxidant phytochemical ingredient for the development and utilization of A. fragrans leaves as a functional supplement (healthy tea) in food and health industry.
- Please conclude according to the main objectives of the research.
Response: Thanks for your suggestion.
Thus, this research afforded a valuable antioxidant phytochemical ingredient for the development and utilization of A. fragrans leaves as a functional supplement (healthy tea) in food and health industry. These results showed that the cytoprotective effect of the isolates was performed on hydrogen peroxide (H2O2)-induced oxidative stress in human liver cancer HepG2 cells. Due to the enrichment of phenolics and remarkable biological activities, A. fragrans could be regarded as a bioactive functional plant and its leave could be used as a potential source of natural antioxidants for food and pharmaceutical applications. This work shed light on the potential application of A. fragrans in the prevention of oxidative stress-induced damage.
Round 2
Reviewer 1 Report
The paper have been improved and now the research is understable.
The research is interesting and a lot of flaws have been corrected
Please in line 46 use "molecules" instead of skeletons.
Author Response
Ref: molecules-1221896
Title: Activity guided isolation of phenolic compositions from Anneslea fragrans Wall. and their cytoprotective effect against hydrogen peroxide induced oxidative stress in HepG2 cells
Reviewer 1:
- Please in line 46 use "molecules" instead of skeletons.
Response: Thank you for your suggestion. The word is revised to “molecules”.
Thank you very much!
Reviewer 2 Report
The paper has been improved. My recommendation is to be published.
Author Response
Ref: molecules-1221896
Title: Activity guided isolation of phenolic compositions from Anneslea fragrans Wall. and their cytoprotective effect against hydrogen peroxide induced oxidative stress in HepG2 cells.
Reviewer 2:
Thank you for your comments. We are truly grateful to yours valuable comments and suggestions. We hope the new manuscript will meet your magazine’s standard.
Best wishes!
Thank you very much!